# Lifestyle Habits Predict Academic Performance in High School Students: The Adolescent Student Academic Performance Longitudinal Study (ASAP)

**DOI:** 10.3390/ijerph17010243

**Published:** 2019-12-29

**Authors:** Marie-Maude Dubuc, Mylène Aubertin-Leheudre, Antony D. Karelis

**Affiliations:** 1Department of Biology, Université du Québec à Montréal, Montreal, QC H3C 3P8, Canada; dubuc.marie-maude@courrier.uqam.ca; 2Department of Exercise Science, Université du Québec à Montréal; Montreal, QC H3C 3P8, Canada; aubertin-leheudre.mylene@uqam.ca

**Keywords:** screen time, eating habits, sleep, physical activity, social media, executive functions, cognition, academic achievement, nutrition

## Abstract

This study aimed to determine if lifestyle habits could predict changes in cognitive control and academic performance in high school students using a longitudinal approach. One hundred and eighty-seven grade seventh to ninth students (mean age: 13.1 ± 1.0 years old) completed a 3-year prospective study. Lifestyle habits, cognitive control, and academic performance were assessed every year during the 3-year study. Results show that in female students, screen time measures were negatively correlated with academic performance and cognitive control. Furthermore, changes (Δs) in sleeping habits were associated with Δs in academic performance in both genders, whereas Δs in eating habits and in studying time were correlated with Δs in academic performance only in male students. Moreover, in female students, screen time, social media use, and eating habits measures seem to predict the variance in the Δs of cognitive control measures (r^2^ between 8.2% and 21.0%), whereas, in male students, studying time, eating, and sleeping habits appear to explain the variance in the Δs of academic performance measures (r^2^ between 5.9% and 24.8%). In conclusion, results of the present study indicate that lifestyle habits were able to predict Δs in cognitive control and academic performance of high school students during a 3-year period.

## 1. Introduction

Adolescence is a developmental period well-known for its physical growth and its brain maturation [1,2], but is also a period often characterized by the prevalence of unhealthy behaviors [3]. Among these unhealthy behaviors often reported are physical inactivity, screen time overuse, skipping breakfast and insufficient sleep [3,4]. For example, Carson et al. [3] observed that, in a sample of 4157 Canadians from 6 to 17 years old, only 17.1% of the participants met all physical activity (60 min per day), screen time (a maximum of 2 h per day), and sleep duration (ranging between 8 to 11 h per day according to the age of the youth) recommendations. The benefits of adopting a healthy lifestyle go further than just improving a health condition. Indeed, during adolescence, less screen time as well as good eating and sleeping habits have been shown to be associated with better cognitive control and academic performance [5,6].

### 1.1. Physical Activity

The practice of physical activity and its impact on students’ academic performance is one of the most documented lifestyle habits to date. Indeed, the results of numerous studies support the idea that the practice of physical activity improves academic performance [7,8,9]. In fact, a panel of experts recently released the Consensus of Copenhagen, which states that the practice of physical activity has a positive effect on academic performance [10]. 

### 1.2. Sleeping Habits

In Canada, a recent analysis of lifestyle habits of 4157 children and adolescents aged between 6 and 17 years old reported that only 24.5% of them follow the guidelines for sleep duration per night (from 9 to 11 h for the 5 to 13 years old and from 8 to 10 h for the 14 to 17 years old) [3]. The lack of sleep that is frequently observed in adolescence seems to be significantly associated with a decline in academic performance [11,12,13]. For example, in a sample of 3120 Americans aged between 13 and 19 years old, Wolfson et Carskadon [14] observed a better academic performance in students who slept on average 34 min more per night and went to bed on average 55 min earlier on weekdays compared to students who did not follow this sleeping pattern. A possible mechanism that could explain the relationship between lack of sleep and academic performance is sleepiness during classes, which could affect academic performance negatively [15]. In addition, lack of sleep may decrease working memory [16] and cognitive functions [6] and in turn, this could lead to a poorer academic performance.

### 1.3. Screen Time

Screen time refers to the time spent in front of a television, computer, tablet, and cellphone for anything other than work or schoolwork. In Canada, 51.8% of children and adolescents appear to follow the recommendations of a maximum of 2 h of screen time per day [3]. The use of screens for a daily duration of more than 2 h may be negatively associated with academic performance [17,18,19]. For example, a study conducted in 395 grade 7 Chileans reported that adolescents who spent more than 2 h per day in front of a screen seems to present a lower academic performance than adolescents who spent 2 h or less per day in front of a screen [17]. Possible mechanisms explaining these associations are the reduction of sleep duration [6] and cognitive control [6,20]. However, results of recent studies suggest that the influence of screen time on academic performance and on cognitive control vary depending on the type of screen used [6]. For example, video games could lead to the improvement of academic engagement [21] and of certain cognitive functions such as visual processing skills [22], while television viewing seems to be associated with a decrease in cognitive control, more specifically in attentional capacities [20]. Moreover, the time of the week during which screen activities are performed seems to affect the relationship between screen time and academic performance differently [18,23,24]. For example, there is evidence to suggest that playing video games on weekdays has a greater negative effect on academic performance than playing video games on weekend [23,24]. However, to our knowledge, no study has examined separately the influence of the time spent in front of different screen types on academic performance.

### 1.4. Eating Habits

The most studied eating habit in students is the daily intake of breakfast. In Canada, 87.9% of adolescents (13 to 17 years old) and 96.9% of children (6 to 12 years old) report eating a daily breakfast [25]. As breakfast is commonly considered as the most important meal of the day [26], its positive impact on academic performance has been reported numerous times [27,28,29]. For example, in Korea, So [28] studied the association between the weekly frequency of breakfast intake and academic performance in 75,643 students from 12 to 18 years old. Results indicated that adolescents who eat breakfast every day have 1.7 to 1.9 more chances to achieve an excellent academic performance than the ones who skip breakfast every day [28]. Other eating habits, including eating quality, have also been identified as being associated with academic performance [30,31]. For example, adolescents with unhealthy diets were less likely to perform well in school than the adolescents with healthy diet in a sample of 395 Chileans students [30]. These relationships between eating habits and academic performance appear to be explained by improved brain function [32,33]. Indeed, it was observed that breakfast was associated with improved neuronal activity [32] as well as with better cognitive control such as working memory and attentional capacities [33]. It has also been reported that a quality diet, evaluated from national recommendations, was associated with better working memory [34]. 

### 1.5. Cognitive Control

As proposed by Hillman et al. [8], cognitive control “is a term used to describe a subset of goal-directed, self-regulatory operations involved in the selection, scheduling, and coordination of computational processes underlying perception, memory, and action”. It should be noted that cognitive control has been established as a determining factor of academic performance [35,36]. More precisely, inhibitory control and working memory are two aspects of cognitive control that have been identified to be strongly related to academic performance [37,38]. For example, in a sample of 51 British children of 11 and 12 years old, working memory was significantly correlated with academic performance in both English and mathematics, whereas inhibitory control was associated with achievement in English, mathematics, and science [37]. However, few studies have investigated, in the same study, the relationships between lifestyle habits with academic performance and cognitive control [39,40]. This multivariate approach may give us a better understanding into a large variety of important factors involved in academic performance.

It should also be noted that most of the existing research on the association between lifestyle habits with cognitive control and academic performance used cross-sectional designs and few were school-based [41], reinforcing the pertinence of performing longitudinal studies directly in high school settings. To our knowledge, no prospective study has examined if the combination of several lifestyle habits such as screen time, studying time, employment status as well as eating and sleeping habits may predict cognitive control and academic performance during adolescence [41,42]. Therefore, the purpose of this study was to determine if lifestyle habits could predict changes (Δs) in cognitive control and academic performance in high school students using a longitudinal approach. Based on the literature [5,6], we hypothesized that a favorable lifestyle habit profile would predict positive Δs in cognitive control and in academic performance. It should be noted that adolescence is a crucial period in which academic performance could determine the probability of being accepted in a specific post-secondary school program. Indeed, there is evidence to suggest that high school students have become very competitive in achieving excellent grades in order to reach their goals [43]. Accordingly, any advantage that could help improve the academic performance of a high school student may be useful. Therefore, gaining a better understanding if lifestyle habits could predict academic performance in high school students has widespread implications for any high school student, educator and policy maker.

## 2. Materials and Methods 

### 2.1. Overview

The results of this study are based on data collected within the Adolescent Student Academic Performance longitudinal project (ASAP), a 3-year prospective study conducted on 205 volunteer adolescents at a single public high school in Montreal, Canada. All participants were in grade 7, 8, or 9 at the beginning of the project. It should be noted that this high school follows a specific educational program called the International Baccalaureate, which corresponds to an elite program in the province of Quebec. All students enrolled into this high school had excellent grades in elementary school and had to achieve an entrance exam prior to their admission. Inclusion criteria were: (1) To be enrolled in grade 7, 8, or 9 in the selected school, (2) to have a normal or corrected-to-normal vision, (3) to be free of attentional disorders or neurological diseases and (4) to be able to complete standard academic performance testing (our primary outcome). All participants and their parents or guardians were fully informed about the nature, goals and protocols of the study and gave their informed consent in writing. All procedures were approved by the school’s administration, by its Governing board, by the school board and by the Ethics Committee of the Faculty of Science at the Université du Québec à Montréal (CÉRPÉ-3-2013-0100A).

### 2.2. Participants 

At the beginning of the study, 205 students were enrolled in the project, which represents 34.8% of the 590 eligible students. Participants and their parents or legal guardians completed screening questionnaires on the health situation of the adolescent in order to confirm certain inclusion criteria. Participants also completed the Child Behavior Check List [44] to screen for any attentional disorders. Thus, 199 participants participated in the study at year 1, with a 1-year follow-up rate of 96% (n = 191) at year 2 and of 98% (n = 187) at year 3 (see Figure 1). At the end, a total of 187 grade seventh to ninth students (mean age: 13.1 ± 1.0 years old at study entry) from the selected high school completed the 3-year follow-up, which represent a 3-year follow-up rate of 94%. The ethnicity of our sample was composed of 115 Caucasians (61%), 34 Asians (18%), 12 Arabs (7%), 8 Hispanics (4%), 6 African-Americans (3%), and 12 mixed (7%).

### 2.3. Demographic Variables

In order to control our analysis for potential cofounding factors, important information on our participants was collected. 

First, the age and the ethnicity of our participants were recorded. Also, the pubertal status was obtained once a year using the Petersen puberty scale [45]. Finally, the socioeconomic status was estimated every year using the Hollingshead four factor index of social status, which is based on marital status, retired/employed status, educational attainment, and occupational title [46]. 

Pubertal status of the participants at baseline: Early pubertal (n = 5; 0 female), mid-pubertal (n = 34; 23 females), late pubertal (n = 24; 1 female), and post-pubertal (n = 124; 92 females). Socioeconomic status of the participants at baseline: Low income (n = 0; 0 female), low-middle income (n = 8; 6 females), middle income (n = 10; 6 females), middle-high income (n = 52; 29 females), and high income (n = 105; 68 females).

### 2.4. Academic Performance

Academic performance was assessed every year using the school’s final report card. In the present study, grades in science, mathematics, language (French) as well as the overall average of each student, in percentage, were reported. The overall average is an overall weighted average calculated by the school using the final grades, in percentage, of all courses taken by a student during the school year. That is, the weighted value of each course used for the overall average was as follows: First language (22%), second language (8%), third language (8%), mathematics (17%), science (17%), history (11%), ethics and religious culture (6%), visual arts (6%), and physical education (6%). It should be noted that all students from the same cohort performed the same language, mathematics and science exams at the exact same time during the school year. Therefore, grades are standardized across all of the students in the different classes.

### 2.5. Cognitive Control

Inhibitory control was assessed once per year using the Flanker task [47]. This cognitive task was performed using a computer and the Inquisit 4.0.9 software [48]. During this task, participants viewed a series of 5 arrows in the middle of the screen inside a rectangle. They had to respond to the directionality (right or left) of the central arrow. This target arrow was surrounded by 4 irrelevant arrows (flankers) that either pointed the same (congruent) or the opposite (incongruent) direction (see Figure 2). Participants had to respond as quickly and accurately as possible pressing keyboard buttons “Q” (on the left side of the keyboard) when the central arrow pointed left and “*p*” (on the right side of the keyboard) when the central arrow pointed right. The task was composed of 2 blocks of 75 trials each, preceded by a block of 10 practice trials (5 congruent and 5 incongruent). During the task, equal numbers of congruent, incongruent, left pointing and right pointing trials were presented to the participants in a random order. Arrows measured 2.5 cm, were black and were presented in a white rectangle for 200 ms with a fixed interstimulus interval of 1700 ms. Total congruent and incongruent accuracy, in percentage, as well as the mean reaction time (MRT) for congruent and incongruent correct answers were collected. In addition, accuracy interference was calculated by subtracting incongruent accuracy from congruent accuracy, and MRT interference was calculated by subtracting congruent MRT from incongruent MRT. Thus, greater interference scores indicate poorer performance. It should be noted that no trimming procedure was conducted. This cognitive task denotes good reliability (Cronbach α = 0.87).

Working memory was assessed once per year using the N-back task [49,50]. This cognitive task was also performed using the Inquisit 4.0.9 software [48]. During this task, 3 cm white letters on a black background were presented once at a time to the students. Participants had to respond as quickly and accurately as possible while fulfilling appropriate conditions (1-back or 2-back) of the task. In the 1-back condition, participants had to press the keyboard button “A” when the current letter was the same as the one presented before (i.e., 1 position back in the sequence), whereas in the 2-back condition, participants had to press the key when the current letter was the same as the one presented before last (i.e., 2 positions back in the sequence; see Figure 3). The task was composed of 3 blocks of 1-back condition and 3 blocks of 2-back condition containing 24 trials each, preceded by 2 blocks (one 1-back and one 2-back) of 10 practice trials each. During the task, block conditions, letters and targets were all presented to the participants in a random order, with a fixed number of 8 targets in each block. Letters were presented during 500 ms with a fixed interstimulus interval of 2500 ms. Premature and late responses as well as false alarms were categorized as errors. Accuracy, in percentage, and the MRT for correct answers in both conditions were collected. Thereafter, the signal-detection parameter (d’) was calculated as z(hit rate)–z(false alarm rate) using the formula provided by Stanislaw and Todorov [51]. This cognitive task denotes good reliability in both conditions (Cronbach α ≥ 0.88).

### 2.6. Lifestyle Habits

Participants self-reported the following lifestyle habits using a questionnaire: Employment status (working or not), studying time, moderate to vigorous physical activity practice, screen time (television, computer and video games, other computer usage and cellphone), time spent on social medias, eating habits (daily number of meals, daily serving of fruits and vegetables and breakfast consumption) and sleeping habits (sleeping schedule, sleep duration, sleep onset latency). Participants had to report all these habits during both the weekdays (WD) and the weekend (WE), except for employment status, studying time, moderate to vigorous physical activity practice, daily number of meals and of serving of fruits and vegetables as well as sleep onset latency. 

### 2.7. Statistical Analysis 

The data are expressed as the mean ± standard deviation (SD). Because interactions for gender were significant in the relationships between lifestyle habits factors with academic performance and cognitive control and because academic performance differs according to gender [52], participants were divided into two groups based on their gender (Female: n = 116; Male: n = 71). Except for the screen time and the time spent on social medias variables, the change (Δ) of all variables between year 1 and year 3, in percentage, was calculated using the following formula:(1)$$\frac{\mathrm{Year}\;3\;\mathrm{value}\;-\;\mathrm{Year}\;1\;\mathrm{value}}{\mathrm{Year}1\;\mathrm{value}}\;\times \;100$$

As some year 1 value were 0 in screen time and time spent on social medias variables, no Δ in percentage was calculated for these variables. Instead, the Δ between year 1 and year 3 in these variables was calculated by subtracting year 1 value to year 3 value. Thereafter, a paired *t*-test was used to compare results between year 1 and 3. Also, a two-proportion z-test was performed to assess proportional differences in employment status between year 1 and year 3. In addition, Pearson’s partial correlations were performed to examine the relation between aspects of lifestyle habits with academic performance and cognitive control in both groups, at baseline and for the Δ. Correlations were controlled for age, pubertal status, socioeconomic status, and ethnicity. Thereafter, a comparison between correlation coefficient values of both genders was performed using the Fisher’s *Z*-transformation [53]. Finally, preliminary analysis showed that, in our sample, associations between cognitive control and academic performance were weak and had a poor ability to predict variations in academic performance during a 3-year period. Therefore, separate linear hierarchical regression analyses were used to identify predictors of Δs in academic performance and in cognitive control measures in both genders. Step 1 analysis included demographic variables that were significantly correlated with measures of academic performance or with measures of cognitive control. Thereafter, measures of lifestyle habits that were significantly correlated with the academic performance or cognitive control measures were included in step 2. It should be noted that the linear regression analysis results were similar when all the covariates were included in the model (Appendix A). Statistical analysis was performed using SPSS 24 for Windows (Chicago, IL, USA). All tests were two-tailed, and significance was defined at *p* < 0.05. 

## 3. Results

Mean academic performance in science, mathematics, language as well as the overall average at baseline were 83.8%, 83.5%, 85.1%, and 85.5%, respectively. Female students had significantly higher grades in science and language as well as higher overall averages than male students. Also, academic performance significantly declined between year 1 and year 3 in both genders (data not shown). In addition, a general improvement in cognitive control measures between year 1 and year 3 was observed in high school students and no difference between genders was found (data not shown). 

Participants working, studying, physical activity and eating habits at year 1 and 3 are shown in Table 1. No changes between year 1 and 3 were noted for all of these habits (except employment status) in both female and male students. 

Screen time habits at year 1 and 3 on WD and WE are presented in Table 2. Both female and male students significantly decreased their television and video games use on WD and WE between year 1 and year 3. However, we noted increases in computer and cellphone use on WE. Moreover, female students spent more time on social media on both WD and WE at year 3 compare to year 1, while in male students this increase was only observed on WE. 

Participants sleeping habits at year 1 and 3 on WD and WE are shown in Table 3. Later bedtime and shorter sleep durations were observed in both female and male students at year 3 compare to year 1 on WD and WE.

Pearson’s partial correlations at baseline between lifestyle habits and academic performance are presented in Table 4. We briefly summarize these relationships since several associations are worth noting. Almost all of the screen time variables were negatively associated with academic performance in female students, whereas very few associations were noted in male students. Moreover, we found several significant correlations between sleep habits and academic performance in female students, however, no significant relationships were observed with male students. In addition, lifestyle habits at baseline in female students presented the following interesting correlations with cognitive control measures: Video games use with 2-back accuracy (WD: *r* = −0.40, *p* = 0.001; WE: *r* = −0.44, *p* < 0.001), with 2-back MRT (WD: *r* = 0.29, *p* = 0.02; WE: *r* = 0.26, *p* = 0.03), and with d’ (WD: *r* = −0.25, *p* = 0.03), social media on WE with d’ (*r* = −0.25, *p* = 0.03), bedtime on WE with Flanker incongruent accuracy (*r* = −0.29, *p* = 0.02), sleep duration on WE with Flanker congruent accuracy (*r* = 0.39, *p* = 0.001), and with 1-back accuracy (*r* = 0.26, *p* = 0.03), and sleep onset latency with Flanker incongruent accuracy (*r* = −0.46, *p* < 0.001). In male students, the following interesting correlations between lifestyle habits at baseline with cognitive control measures were found: Television use on WE with Flanker MRT interference (*r* = 0.46, *p* = 0.001) and with d’ (*r* = −0.029, *p* = 0.04) as well as sleep onset latency with Flanker MRT interference (*r* = 0.59, *p* < 0.001) and with 2-back accuracy (*r* = −0.32, *p* = 0.03).

Furthermore, Pearson’s partial correlations between lifestyle habits and Δs in academic performance showed that, in female students, Δ in video games use on WE was associated with Δ in language (*r* = −0.25, *p* = 0.04) and that Δ in sleep duration on WE was related with Δ in overall average (*r* = 0.35, *p* = 0.004) and with Δ in language (*r* = 0.38, *p* = 0.002). Also, Δ in bedtime on WE and Δ in sleep duration on WE were both correlated with Δ in Flanker congruent accuracy (*r* = −0.36, *p* = 0.003; *r* = 0.25, *p* = 0.049, respectively). In addition, social media use at baseline was associated with Δ in Flanker congruent MRT (*r* = 0.28, *p* = 0.24) and with Δ in 2-back MRT (*r* = 0.37, *p* = 0.003), and number of daily meals at baseline was related to Δ in both Flanker congruent and incongruent MRT (*r* = −0.31, *p* = 0.01; *r* = −0.29, *p* = 0.02, respectively) and to Δ in 1-back accuracy (*r* = 0.40, *p* = 0.001). In male students, Δ in studying time was related to Δ in science (*r* = 0.38, *p* = 0.01), while Δ in daily servings of fruits and vegetables was associated with Δ in overall average and in science (*r* = 0.43, *p* = 0.004; *r* = 0.34, *p* = 0.03, respectively). In addition, Δ in cellphone use on WE, Δ in bedtime on WD and WE and Δ in sleep duration on WD were all significantly correlated with Δ in mathematics (*r* = −0.30, *p* = 0.046; *r* = −0.49, *p* = 0.001; *r* = −0.39, *p* = 0.01; *r* = 0.38, *p* = 0.01, respectively). Moreover, Δ in computer use on WE, daily servings of fruits and vegetables at baseline, bedtime and sleep duration on WD at baseline were all significantly correlated with Δ in Flanker MRT interference (*r* = 0.34, *p* = 0.03; *r* = −0.41, *p* = 0.007; *r* = 0.36, *p* = 0.02; *r* = −0.30, *p* = 0.048, respectively). Also, Δ in social media use on WE was associated with Δ in 2-back accuracy (*r* = −0.35, *p* = 0.02) and Δ in sleep duration on WD was correlated with Δ in Flanker congruent accuracy (*r* = −0.30, *p* = 0.05; *r* = 0.33, *p* = 0.03, respectively). Furthermore, bedtime on WD at baseline was associated with Δ in Flanker incongruent accuracy (*r* = −0.38, *p* = 0.01; *r* = −0.30, *p* = 0.049; *r* = −0.33, *p* = 0.03, respectively).

Finally, linear regression analyses showed that, in female students, no lifestyle habits factors could explain the variance in the Δs in overall average, science and mathematics, whereas Δ in wake-up time on WE explained 6.2% of the variance in the Δ in language (*p* = 0.03). However, lifestyle habits factors could explain between 8.2% and 21.0% of the variance in the Δs in cognitive control measures (see Table 5). In male students, lifestyle habits factors could predict between 5.9% and 24.8% of the variance in the Δs in academic performance measures (see Table 6). Moreover, Δ in social media use on WE predicted 8.1% of the variance in Δ 2-back accuracy (*p* = 0.02). Demographic variables did not explain variations in academic performance and cognitive control in female and male students, except for Δs in science and in mathematics in male students, where age was an important predictor of variance (see Table 6).

## 4. Discussion

It was hypothesized that a favorable lifestyle habits profile would predict positive Δs in cognitive control and academic performance in high school students. Based on our results, this hypothesis was confirmed. 

First, we observed significant correlations between lifestyle habits with cognitive control and academic performance in high school students. For example, in female students, screen time measures were negatively correlated with academic performance (with *r* values up to 0.44) and cognitive control (with *r* values up to 0.44). These
findings are in line with results from previous studies [6,54]. For example, Kantonomaa et al. [54] observed that viewing television less than one hour per day and using computer and video games less than one hour per day were both associated with a higher academic performance in a sample of 8061 adolescents of 15 and 16 years old. In contrast, a study conducted in 371 undergraduate students reported that screen time on both WD and WE was not related with working memory, however, more than 3 h of screen time per day on WD was negatively associated with academic performance [55]. Furthermore, Δs in sleeping habits were associated with Δs in academic performance in both genders (with *r* values up to 0.49), whereas Δs in eating habits and in studying time were correlated with Δs in academic performance only in male students (with *r* values up to 0.43 and *r* = 0.38, respectively). Similar general findings were reported in a sample of 4625 adolescents from 14 to 18 years old from the United States [56]. In that study, the authors examined the relationships between sleep duration and diet with academic achievement and found that eating a daily breakfast and consuming salads weekly were associated with a better academic performance [56]. However, they observed no association between sleeping at least eight hours per night and academic performance [56]. 

In the present study, we attempted to develop a model that includes multiple sociodemographic and lifestyle habits measurements that might help us better understand the predictors of cognitive control and academic performance. Results from the linear hierarchical regression analyses showed that, in female students, sleeping habits seems to weakly explain the variance in the Δ in language (r^2^ = 0.062), while screen time, social media use and eating habits measures seem to predict the variance in the Δ of cognitive control measures (r^2^ between 8.2% and 21.0%). In male students, studying time, eating and sleeping habits appear to explain the variance in the Δ of academic performance measures (r^2^ between 5.9% and 24.8%), whereas screen time and social media use seem to predict the variance in the Δ of cognitive control measures (r^2^ between 8.6% and 14.0%). Taken together, results of the present study suggest that the relationships between lifestyle habits with cognitive control and academic performance differ between genders. Thus, our results indicate that it may be important to analyze male and female students separately when studying academic performance or cognitive control in order to detect potential disparities that could exist between female and male students. 

Furthermore, the findings in the present study are mostly in line with results from a previous 1-year prospective study [42]. In that study, the authors examined if a combination of lifestyle habits, including screen time as well as eating and sleeping habits, could impact academic performance of 4253 elementary students from 10 and 11 years old [42]. They concluded that adherence to screen time as well as eating and sleeping recommendations increase the likelihood of having a better academic performance in mathematics, reading and writing. Also, they reported that adherence to multiple healthy lifestyle recommendations have an additive effect on the associations with academic performance [42]. In contrast, results of a 2-year longitudinal study conducted in the United Kingdom in 11014 children indicated that screen time at 5 years old did not predict variations in attention capacity 2 years later [57].

There were some limitations to the present study. First, our findings are limited to a population of higher than average academic performers students from a single French-Canadian public high school in Montreal, Canada. Nonetheless, our results are strengthened by studying a homogenous population. Second, despite reducing our sample size, separating our sample into two groups allows us to gain a better understanding of the factors associated with academic performance in both female and male students. In turn, this may lead to the planning of better interventions specifically targeting female or male students in order to improve their academic performance. Moreover, due to the differences in academic curriculums and assessments in high schools all around the world, it is difficult to establish comparisons in academic performance with other investigations. However, we used grades in percentage to facilitate comparisons and to allow conversions to letter grades systems. In addition, the intelligence quotient was not measured in the present study. Finally, due to a logistic reality, another limitation was the use of self-reported measures of lifestyle habits. Despite these limitations, our results are strengthened by using a longitudinal approach. 

## 5. Conclusions

In conclusion, results of the present study indicate that lifestyle habits were able to predict Δs in cognitive control and academic performance of high school students during a 3-year period. Therefore, high school policy makers and school educators could consider adopting and applying politics to promote the importance of following a healthy lifestyle by developing effective intervention programs, which may lead to better academic performances. Specifically, high school educators may consider developing new courses that will help educate students in adopting a healthy lifestyle. In addition, educators could plan a great variety of free intervention programs to students by organizing workshops and afterschool programs that promote the importance of improving eating habits (e.g., increasing the consumption of fruits/vegetables), sleep habits (early bedtime), and lessening screen time.

## Figures and Tables

**Figure 1 ijerph-17-00243-f001:**
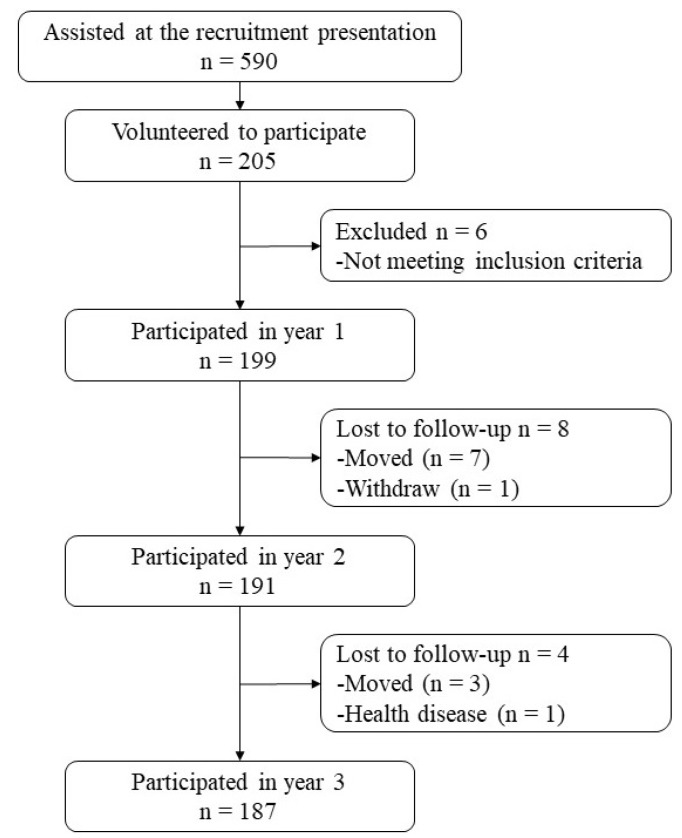
Participants flowchart.

**Figure 2 ijerph-17-00243-f002:**
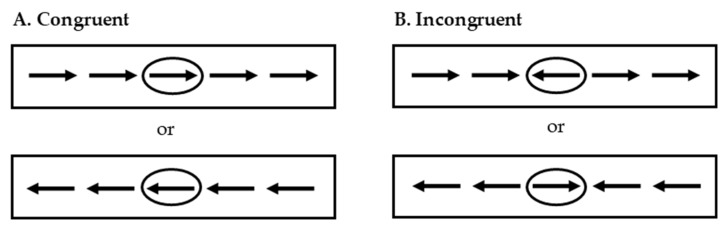
Flanker task. (**A**) Presented the two possibilities for a congruent trial. (**B**) Presented the two possibilities for an incongruent trial.

**Figure 3 ijerph-17-00243-f003:**
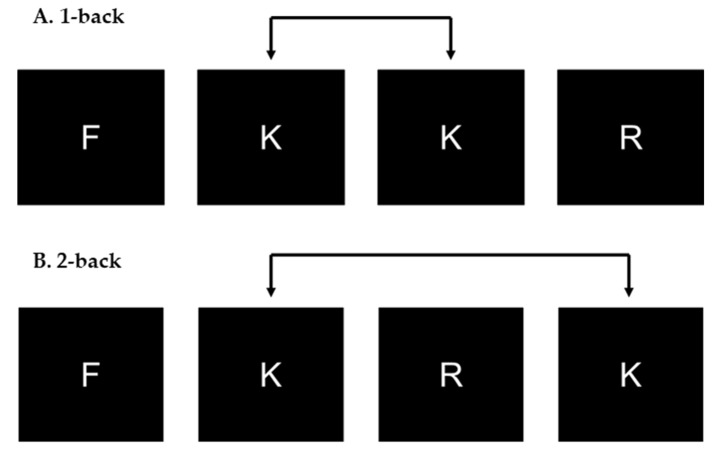
N-back task. (**A**) Presented a 1-back condition trial. (**B**) Presented a 2-back condition trial.

**Table 1 ijerph-17-00243-t001:** Working, studying, physical activity, and eating habits of high school female and male students.

Variables	Female Students n = 116	Male Students n = 71
Students working (n (%) workers), *year 1*	18 (16.1)	12 (17.4)
Students working (n (%) workers), *year 3*	35 (30.2) *	10 (14.1)
Studying time (h/week), *year 1*	11.4 ± 8.4 (1–50)	9.5 ± 7.3 (1–36)
Studying time (h/week), *year 3*	12.6 ± 9.7 (0–48)	10.9 ± 9.9 (0–48)
Physical activity (h/week), *year 1*	6.2 ± 5.5 (0–30)	6.7 ± 5.4 (0–30)
Physical activity (h/week), *year 3*	5.2 ± 5.1 ^†^ (0–30)	6.7 ± 5.0 (0–22)
Number of meals/day, *year 1*	3.0 ± 0.6 (1–6)	3.1 ± 0.5 (2–5)
Number of meals/day, *year 3*	3.0 ± 0.6 (1–5)	3.3 ± 0.6 ^†^ (2–5)
Serving of fruits and vegetables/day, *year 1*	4.1 ± 1.6 (1–10)	4.0 ± 1.8 (1–10)
Serving of fruits and vegetables/day, *year 3*	4.1 ± 1.6 (1–8)	4.4 ± 2.3 ^†^ (0–10)
Breakfast consumers on weekdays (n (%)), *year 1*	96 (83.5)	63 (88.7)
Breakfast consumers on weekdays (n (%)), *year 3*	91 (79.1)	64 (90.1)
Breakfast consumers on weekend (n (%)), *year 1*	103 (89.6)	66 (94.3)
Breakfast consumers on weekend (n (%)), *year 3*	101 (87.8)	66 (93.0)

Values are mean ± standard deviation (range). * Significantly different from year 1. ^†^ Represents a tendency (*p* between 0.05 and 0.08) in its difference from year 1.

**Table 2 ijerph-17-00243-t002:** Screen time habits on weekdays and weekends of high school female and male students.

Variables	Female Students n = 116		Male Students n = 71	
	Weekdays	Weekend	Weekdays	Weekend
Screen time (h/day)				
Television, *year 1*	1.6 ± 1.5 (0–7)	2.8 ± 2.0 (0–10)	1.2 ± 1.1 (0–6)	2.6 ± 2.0 (0–8)
Television, *year 3*	1.2 ± 1.3 * (0–5)	2.4 ± 1.9 ^†^ (0–8)	0.9 ± 1.0 * (0–6)	2.1 ± 1.8 * (0–8)
Computer, *year 1*	1.7 ± 1.5 (0–7)	2.3 ± 2.2 (0–13)	2.2 ± 1.9 (0–10)	2.3 ± 2.2 (0–10)
Computer, *year 3*	1.6 ± 1.6 (0–6)	2.8 ± 2.9 * (0–11)	1.7 ± 1.7 (0–7)	3.1 ± 2.6 * (0–14)
Video games, *year 1*	0.7 ± 1.5 (0–7)	1.2 ± 1.9 (0–8)	1.4 ± 1.8 (0–7)	2.7 ± 2.2 (0–8)
Video games, *year 3*	0.2 ± 0.6 * (0–4)	0.4 ± 1.0 * (0–6)	0.7 ± 1.1 * (0–6)	1.9 ± 1.8 * (0–8)
Cellphone, *year1*	1.1 ± 1.6 (0–9)	1.6 ± 2.4 (0–10)	0.9 ± 1.7 (0–10)	1.1 ± 1.9 (0–8)
Cellphone, *year 3*	1.9 ± 2.6 (0–15)	2.7 ± 3.1 * (0–14)	1.4 ± 2.1 (0–10)	2.0 ± 2.8 * (0–12)
Social media use, *year 1*	1.9 ± 1.8 (0–8)	2.9 ± 2.9 (0–15)	1.5 ± 1.6 (0–9)	1.6 ± 1.9 (0–8)
Social media use, *year 3*	2.5 ± 2.9 * (0–15)	3.6 ± 3.2 * (0–15)	1.5 ± 1.9 (0–10)	2.2 ± 2.3 * (0–12)

Values are mean ± standard deviation (range). * Significantly different from year 1. ^†^ Represents a tendency (*p* = 0.066) in its difference from year 1.

**Table 3 ijerph-17-00243-t003:** Sleeping habits on weekdays and weekends of high school female and male students.

Variables	Female students n = 116		Male students n = 71	
	Weekdays	Weekend	Weekdays	Weekend
Bedtime ^§^, *year 1*	1.9 ± 0.9 (0–5)	3.0 ± 1.3 (0–8)	1.6 ± 0.9 (0–4)	2.8 ± 1.2 (0–6)
Bedtime ^§^, *year 3*	2.5 ± 1.0 * (0–5)	3.4 ± 1.2 * (0–7)	2.4 ± 1.0 * (0–5)	3.5 ± 1.4 * (0–5)
Wake-up time (AM), *year 1*	6.3 ± 0.5 (5–7)	9.2 ± 1.5 (6–13)	6.2 ± 0.5 (5–7)	8.8 ± 1.4 (6–12)
Wake-up time (AM), *year 3*	6.3 ± 0.6 (5–7)	9.3 ± 1.5 (5–12)	6.3 ± 0.5 * (5–7)	9.0 ± 1.4 ^†^ (6–12)
Sleep duration (h), *year 1*	8.4 ± 0.9 (5–10)	10.2 ± 1.5 (5–13)	8.6 ± 0.9 (6–10)	10.0 ± 1.3 (7–14)
Sleep duration (h), *year 3*	7.8 ± 1.1 * (4–10)	10.0 ± 1.2 * (5–12)	7.9 ± 1.0 * (5–10)	9.5 ± 1.6 * (5–13)
Sleep onset latency (min), *year1*	24.7 ± 25.8 (0–180)		18.3 ± 18.5 (0–120)	
Sleep onset latency (min), *year 3*	24.5 ± 27.9 (0–180)		17.6 ± 16.1 (0–90)	

Values are mean ± standard deviation (SD). ^§^ Bedtime is represented by the number of hours past 8 pm (e.g., 8 pm = 0; 9 pm = 1; 1 am = 5; etc.). * Significantly different from year 1. ^†^ Represents a tendency (*p* = 0.050) in its difference from year 1.

**Table 4 ijerph-17-00243-t004:** Correlations between lifestyle habits with academic performance measures at baseline in high school female and male students.

	Female students				Male students			
	OA	SCI	MAT	LAN	OA	SCI	MAT	LAN
Studying time	0.00	−0.10	−0.02	−0.01	0.29 *	0.09	0.18	0.28 ^†^
Physical activity	0.00	−0.12	−0.05	0.10	0.08	0.07	0.09	−0.05
Number of meals/day	0.11	0.11	0.15	0.17	−0.21 ^§^	0.01	−0.25 ^§^	−0.15 ^§^
Serving of fruits and vegetables/day	0.06	−0.06	0.08	0.03	0.25	0.13	0.24	0.17
Screen usage								
Television WD	−0.34 **	−0.19	−0.36 **	−0.37 **	−0.01 ^§^	−0.09	−0.04 ^§^	0.01 ^§^
Television WE	−0.33 **	−0.16	−0.32 **	−0.29 *	−0.10	−0.10	−0.15	−0.13
Computer WD	−0.17	−0.01	−0.14	−0.25 *	−0.01	−0.10	0.04	0.01
Computer WE	0.07	0.08	0.04	0.06	0.16	0.13	0.20	0.03
Video games WD	−0.17	−0.17	−0.16	−0.23 ^†^	−0.04	−0.15	0.02	−0.08
Video games WE	−0.19	−0.05	−0.22 ^†^	−0.25 *	−0.31 *	−0.27 ^†^	−0.32 *	−0.17
Cellphone WD	−0.26 *	−0.29 *	−0.35 **	−0.36 **	0.22 ^§^	0.07 ^§^	0.20 ^§^	0.09 ^§^
Cellphone WE	−0.33 **	−0.25 *	−0.39 **	−0.42 **	0.25 ^§^	0.24 ^§^	0.14 ^§^	0.16 ^§^
Social media WD	−0.02	0.03	−0.03	−0.17	0.00	−0.09	0.03	−0.02
Social media WE	−0.24 *	−0.19	−0.21	−0.37 **	0.03	0.03	0.00	−0.03 ^§^
Sleep habits								
Bedtime WD	0.11	0.23 ^†^	0.01	0.22	−0.08	−0.05	−0.11	−0.19 ^§^
Bedtime WE	−0.26 *	−0.17	−0.25 *	−0.30 *	−0.08	0.04	−0.12	−0.04 ^§^
Wake up time WD	0.11	0.09	0.05	0.10	0.06	0.10	0.12	0.01
Wake up time WE	0.06	0.11	0.00	−0.01	−0.03	−0.12	−0.17	0.03
Sleep duration WD	−0.05	−0.18	0.01	0.04	0.11	0.10	0.17	0.20
Sleep duration WE	0.28 *	0.26 *	0.20	0.23 ^†^	0.04	−0.15 ^§^	−0.06	0.07
Sleep onset latency	−0.22 ^†^	−0.12	−0.26 *	−0.22 ^†^	−0.22	−0.16	−0.26 ^†^	−0.24

^†^ Tendency (0.05 < *p* < 0.08), * *p* < 0.05, ** *p* < 0.01. ^§^ Significantly different from female students (*p* < 0.05). OA: Overall average, SCI: Science, MAT: Mathematics, LAN: Language, WD: Weekdays, WE: Weekend. Control variables: Age, pubertal status, socioeconomic status, and ethnicity.

**Table 5 ijerph-17-00243-t005:** Hierarchical regression analysis regarding independent predictors of cognitive control in high school female students.

Dependent Variables	Independent Variables	*β*	Total r^2^	*p* Value
ΔFlanker congruent MRT	Social media on WD at Y1	0.28	0.147	0.001
	Daily meals at Y1	−0.29		
ΔFlanker incongruent MRT	Daily meals at Y1	−0.24	0.099	0.012
	Social media on WD at Y1	0.22		
Δ1-back accuracy	Daily meals at Y1	0.39	0.210	0.000
	ΔTotal screen on WD	−0.24		
Δ2-back accuracy	ΔVideo games on WE	−0.32	0.100	0.007
Δ2-back MRT	Social media on WD at Y1	0.41	0.206	0.000
	Daily meals at Y1	−0.25		

MRT: Mean reaction time, WD: weekdays, WE: Weekend, Y1: Year 1. Independent predictors included in the model varied between dependent variables based on significant correlations. None of the demographics variables (age, pubertal status and socioeconomic status) included in step 1 analysis remained in the final model.

**Table 6 ijerph-17-00243-t006:** Hierarchical regression analysis regarding independent predictors of academic performance in high school male students.

Dependent Variables	Independent Variables	*β*	Total r^2^	*p* Value
ΔOverall average	ΔDaily servings of F/V	0.50	0.248	0.000
ΔScience	Age	0.60	0.392	0.000
	ΔStudying time	0.24	0.059	0.009
ΔMathematics	Age	−0.26	0.069	0.027
	ΔBedtime on WD	−0.45	0.202	0.000
ΔLanguage	ΔBreakfast on WE	−0.25	0.064	0.059

F/V: Fruits and vegetables, WD: Weekdays, WE: Weekend. Independent predictors included in the model varied between dependent variables based on significant correlations. None of the demographics variables (age, pubertal status and socioeconomic status) included in step 1 analysis remained in the final model for Δ in overall average and Δ in in language.

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
