# Peer review of "Lifestyle Habits Predict Academic Performance in High School Students: The Adolescent Student Academic Performance Longitudinal Study (ASAP)"

_ijerph, 2019, doi:10.3390/ijerph17010243_

Round 1

Reviewer 1 Report

Introduction section is a bit lengthy; it can be summarized concisely.
I recommend to make tables in a standard manner.
Typos need to be corrected.

Author Response

COMMENTS AND SUGGESTIONS FOR AUTHORS

Comment 1: Introduction section is a bit lengthy; it can be summarized concisely.

Answer 1: We understand that the reviewer finds the introduction section a bit lengthy. However, in the previous comments received, it was mentioned by Reviewer 3 that “the literature review was too short and limited”. Specific demands from Reviewers 2 and 3 to “clearly define in the introduction what is cognitive control”, “to have separate detailed section for screen time, eating habits and sleeping habits” and to “provide more literature reviews regarding the possibility of different implications of weekend and weekdays habits” were made. Therefore, we are reluctant to make any further changes to the introduction section in order to avoid confusion with the other reviewers.

Comment 2: I recommend to make tables in a standard manner.

Answer 2: Please note that we followed the journal’s instructions on how to create tables. Therefore, we leave to the discretion of the editorial office if specific changes to the tables need to be made.

Comment 3: Typos need to be corrected. 

Answer 3: We carefully revised the manuscript to ensure that no typos remained

Reviewer 2 Report

1) Table 1 - Ligne 278: 0.05 instead of o.05

2) The authors should report the signal-detection parameter. In my humble opinion, this is the only way to erase the random answers given by the participants when performing the n-back task. Certainly, the percentage of correct answers is accepted in the literature. But among the right answers, there may be random answers. Indeed, during n-back, the probability of giving a correct answer is the same as that of giving a wrong answer. This probability is 0.5, hence the importance of reporting the signal-detection parameter.

Author Response

COMMENTS AND SUGGESTIONS FOR AUTHORS

 Comment 1: Table 1 - Line 278: 0.05 instead of o.05

Answer 1: This was corrected.

Comment 2: The authors should report the signal-detection parameter. In my humble opinion, this is the only way to erase the random answers given by the participants when performing the n-back task. Certainly, the percentage of correct answers is accepted in the literature. But among the right answers, there may be random answers. Indeed, during n-back, the probability of giving a correct answer is the same as that of giving a wrong answer. This probability is 0.5, hence the importance of reporting the signal-detection parameter.

Answer 2: As suggested by the reviewer, we have now reported the signal-detection parameter. It is now mentioned in the methods section (page 6, lines 222-223): “Thereafter, the signal-detection parameter (d’) was calculated as z(hit rate) – z(false alarm rate) using the formula provided by Stanislaw and Todorov [52].” We have also adjusted the results section accordingly.

Reviewer 3 Report

The paper has significantly improved and I appreciate the authors' revision. The paper is almost ready for publication. I have a few minor comments that require the authors' attention:

1. I understand that there is no trimming for flanker task. In that case, the authors should report in the manuscript that the trimming procedure was not conducted

2. The authors mentioned that the findings from the analyses with theoretically-driven covariates were the same as the results shown in the present manuscript. If that is the case, the authors must report it in the main text that the results did not differ when all the covariates were included in the model. Perhaps, it will be good for the authors to include the analyses in the appendix. That will provide clearer for the future readers.

Author Response

COMMENTS AND SUGGESTIONS FOR AUTHORS

General comment: The paper has significantly improved and I appreciate the authors' revision. The paper is almost ready for publication. I have a few minor comments that require the authors' attention:

Answer: We thank the reviewer for this comment.

Comment 1: I understand that there is no trimming for flanker task. In that case, the authors should report in the manuscript that the trimming procedure was not conducted.

Answer 1: This was now reported in the manuscript (page 5, lines 202-203): “It should be noted that no trimming procedure was conducted.”

Comment 2: The authors mentioned that the findings from the analyses with theoretically-driven covariates were the same as the results shown in the present manuscript. If that is the case, the authors must report it in the main text that the results did not differ when all the covariates were included in the model. Perhaps, it will be good for the authors to include the analyses in the appendix. That will provide clearer for the future readers.

Answer 2: We have now reported that the findings from the analyses with theoretically-driven covariates were the same as the results shown in the present manuscript (page 7, lines 264-265): “It should be noted that the linear regression analysis results were similar when all the covariates were included in the model (Supplementary file: Appendix 1).”

This manuscript is a resubmission of an earlier submission. The following is a list of the peer review reports and author responses from that submission.

Round 1

Reviewer 1 Report

This is a longitudinal study on the association between individual lifestyle factors with changes in academic performance among 187 students (grade 7 to 9) in Canada.

2.2 Participants

- Is the participating school private school or public school? Please clarify. It seems national (public) school.

- Figure 1 indicates that 590 students were eligible. However, this information is not included in Participants section.

- Also, please clarify the follow-up rate as well as rate of completion of the 3-yr follow-up exam in text.

- I am not sure whether it is valid to mix different grade children. Please justify using existing literature.

- In addition, grade was not considered in the multivariable-adjusted models.

2.3 Demographic variables

- Did the authors obtain data on socioeconomic factors during follow-up?

2.4 Academic performance

- I could not understand the validity of the measurement for academic performance. This is main outcome. Therefore, this point should be explained in more detail, including masking of outcome assessment.

- Rationale is needed to combine science, mathematics, and language.

- Are there any other domains of academic performance?

- Please clarify how the authors weighed the overall average performance.

2.6 Lifestyle habits

- Please explain the measure of social media use in more detail.

2.7 Statistical analysis

- The authors performed sex-specific analysis. Did the authors observe any effect modification by sex on the association between lifestyle factors and academic performance?

- As the range or distribution of the outcomes is not shown, it is difficult to judge the validity of the analysis.

- Is the p-value one-sided or two-sided?

- The results are difficult to interpret. Showing standardized beta coefficient may help understand the results.

Reviewer 2 Report

This This study aimed to determine if lifestyle habits could predict changes in cognitive control and academic performance in high school students using a longitudinal approach. After following a longitudinal protocol for three years, the results showed that in female students, screen time measures were negatively correlated with academic performance and cognitive control. Furthermore, changes (Δs) in sleeping habits were associated with Δs in academic performance in both genders, whereas Δs in eating habits and in studying time were correlated with Δs in academic performance only in male students. Moreover, in female students, screen time, social media use and eating habits measures seem to predict the variance in the Δs of cognitive control measures (r2 between 8.2% and 21.0%), whereas, in male students, studying time, eating and sleeping habits appear to explain the variance in the Δs of academic performance measures (r2 between 5.9% and 24.8%). The idea is laudable, and the results are encouraging, but the authors must rework their article on some points in order to improve it. My comments are listed below.

The authors must clearly define in the introduction what is "cognitive control". Indeed, cognitive control is a variable of interest in the experimental protocol. The authors must also always justify in the introduction why they took inhibitory control and working memory and not cognitive flexibility. In short, the authors must write a paragraph to clarify all this.

Lines 100-104: In my opinion, the authors should control the level of physical activity or CRF because several studies have shown that this factor affects executive and academic performance. Another variable that is also missing from this study is the intelligence quotient since it correlates strongly with academic performance. However, in this study the authors did not perform this measurement. It therefore seems important to me that the authors highlight these limitations in the discussion.

Lines 119-135: The authors must make a figure for Flanker's task.

Lines 126-127: For the flanker task, the authors used a block of 10 trials. In my opinion, this familiarization phase is for familiarization. The authors must therefore justify why they chose a short familiarization block.

The authors did not specify how they measured the accuracy. For the n-back task, accuracy is measured by the formula of Stanislaw & Todorov 1999. Authors must use this formula. (Stanislaw, H.; Todorov, N. Calculation of signal detection theory measures. Behav. Res. Methods Instrum. Comput. 1999, 31, 137–149 ; Agbangla, N.F.; Audiffren, M.; Pylouster, J.; Albinet, C.T. Working Memory, Cognitive Load and Cardiorespiratory Fitness: Testing the CRUNCH Model with Near-Infrared Spectroscopy. Brain Sci. 2019, 9, 38.)

Reviewer 3 Report

This is an interesting and promising paper. I feel that the paper will provide important contribution to the field. However, I have several major concerns that require the authors' attention. As a result, the paper is not ready for publication in its current state. I hope that the authors can address these concerns seriously:

1. There are many analyses that were conducted in the current study (It is important to note that the authors focused on different specific type of screen time, eating and sleep habits) but the literature review was too short and limited. A more throughout literature review is necessary especially since the evidence regarding screen time and academic performance are still inconclusive. It will be good for the authors to have separate detailed section for screen time, eating habits and sleeping habits. Example of some relevant articles:

Esteban-Cornejo, I., Martinez-Gomez, D., Sallis, J. F., Cabanas-Sánchez, V., Fernández-Santos, J., Castro-Piñero, J., ... & UP & DOWN Study Group. (2015). Objectively measured and self-reported leisure-time sedentary behavior and academic performance in youth: The UP&DOWN Study. Preventive Medicine, 77, 106-111.

Przybylski, A. K., & Mishkin, A. F. (2016). How the quantity and quality of electronic gaming relates to adolescents’ academic engagement and psychosocial adjustment. Psychology of Popular Media Culture, 5(2), 145.

Ruiz, J. R., Ortega, F. B., Castillo, R., Martín-Matillas, M., Kwak, L., Vicente-Rodríguez, G., ... & AVENA Study Group. (2010). Physical activity, fitness, weight status, and cognitive performance in adolescents. The Journal of pediatrics, 157(6), 917-922.

2. As most of the analyses in the this paper separate weekend and weekdays habits, it is important for the authors to discuss or provide more literature reviews regarding the possibility of different implications of weekend and weekdays habits.

For example, in terms of academic performances, there are some evidence that the negative relationship between video gaming and academic performance is stronger on weekdays than weekends. Some relevant articles:

Drummond, A., & Sauer, J. D. (2020). Timesplitters: Playing video games before (but not after) school on weekdays is associated with poorer adolescent academic performance. A test of competing theoretical accounts. Computers & Education, 144, 103704.

Hartanto, A., Toh, W. X., & Yang, H. (2018). Context counts: The different implications of weekday and weekend video gaming for academic performance in mathematics, reading, and science. Computers & Education, 120, 51-63.

3. How the income is categorized should be elaborated in the method section

4. In the academic performance section, more information about how the academic performance is measured is necessary. What is the final report card consists? It is all exam based? Is it standardized across the students in different classes? This will have important implications on how the results can be interpreted.

5. It will be good to report the reliability of the cognitive task. The author can calculate it using odd-even method.

6. Did the authors trim the outliers in the flanker task? More information is necessary

7. I feel that a graphical representation of the cognitive tasks will be very helpful. Please consider to provide a figure regarding the procedure of the cognitive task.

8. I am very concerned about the statistical technique employed to analyze the data. I feel that the formula used to calculate changes is not that appropriate. More justification is necessary. A better way to analyze the current data is by using cross-lagged panel analysis. Here, the authors will be able to examine the changes and examine the bidirectional association between lifestyle habits and academic performance/cognitive ability 

9. It is not clear why there were separated by gender. A more detailed elaboration is necessary. If gender is an important consideration, the authors should examine whether the relations between lifestyle habits and academic performance was significantly moderated by gender. It is important to note that the sample size was too small when separate analyses were conducted. Something that the authors should acknowledge.

10. I also have a serious concern regarding the hierarchical regression analyses that were conducted. The analyses were flawed because the predictors included in the model were based on significant correlations. This is not appropriate because it will inflate the coefficient of the independent variable. Theoretically-driven covariates should be included regardless of their significance to provide accurate coefficient of the main independent variable 
